# Unveiling the Potential of BenzylethyleneAryl–Urea Scaffolds for the Design of New Onco Immunomodulating Agents

**DOI:** 10.3390/ph16060808

**Published:** 2023-05-29

**Authors:** Raquel Gil-Edo, Santiago Royo, Miguel Carda, Eva Falomir

**Affiliations:** 1Inorganic and Organic Chemistry Department, University Jaume I, 12071 Castellón, Spain; ragil@uji.es (R.G.-E.); mcarda@uji.es (M.C.); 2Institute of Agronomic Engineering for Development, Polytechnic University of Valencia, 46022 Valencia, Spain; sroyo@uji.es

**Keywords:** tumor microenvironment, aryl urea, angiogenesis, VEGFR-2, PD-L1, immune checkpoints, CD-11b, CD-80, THP-1, HT-29, co-cultures

## Abstract

This work focuses on the development of thirteen benzylethylenearyl ureas and one carbamate. After the synthesis and purification of the compounds, we studied their antiproliferative action on cell lines, such as HEK-293, and cancer ones, such as HT-29, MCF-7 or A-549, on the immune Jurkat T-cells and endothelial cells HMEC-1. Compounds **C.1**, **C.3**, **C.12** and **C.14** were selected for further biological studies to establish their potential as immunomodulating agents. Some of the derivatives exhibited significant inhibitory effects on both targets: PD-L1 and VEGFR-2 in the HT-29 cell line, showing that urea **C.12** is active against both targets. Some compounds could inhibit more than 50% of cancer cell proliferation compared to non-treated ones when assessed in co-cultures using HT-29 and THP-1 cells. In addition, they significantly reduced CD11b expression, which is a promising target for immune modulation in anticancer immunotherapies.

## 1. Introduction

Over the last fifty years, significant effort has been invested in cancer research to uncover factors that may cause cancer and find more and better treatments and improve cancer patients’ quality of life. One of the difficulties in the treatment of this disease is the large number of biological processes that are involved in both the genesis and the development of tumor growth [1]. Basic research to unravel all the processes involved in tumor growth and expansion is still very much needed to advance the development of new and better therapeutic applications [2]; in addition, the relevant role that the tumor microenvironment (TME) plays in both tumorogenesis and metastatogenesis has recently been confirmed [3]. Therefore, therapies targeting components of the tumor microenvironment, in addition to cancer cells, could become an excellent anticancer treatment. It has been shown that immune cells, such as macrophages, and biological processes, such as angiogenesis and inflammation, are directly responsible for shaping and favoring the development of TME and, with it, tumor spread. For this reason, the tumor microenvironment is increasingly being considered a complex biological target that may enable the development of more novel anticancer therapies [4].

Endothelial cells within the TME are actively involved in so-called tumor angiogenesis [5], which allows the creation of a network of blood microvessels that maintain and enable tumor progression [6]. On the other hand, tumor-associated macrophages [7] can enhance cancer invasion, not only in primary but also in metastatic regions, through the promotion of basement membrane degradation and deposition, angiogenesis, leukocyte recruitment, and general immune suppression [8]. In fact, it is clinically demonstrated that antiangiogenic therapies can modulate and remodel TME resulting in increased efficacy of immunotherapies [9]. For example, antiangiogenic treatments can reduce the number of suppressive immune cells and increase the number of immune-stimulating cells in TME, which can enhance the response to immunotherapy [10]. Additionally, antiangiogenic treatments can increase the expression of antigens on cancer cells, making them more visible to the immune system and easier to target. For this reason, combining immunotherapies with antiangiogenic treatments has shown promise in improving treatment outcomes in several types of cancer, including renal cell carcinoma, lung cancer and colorectal cancer. In some cases, this combination approach has led to longer progression-free survival and overall survival rates than either approach alone. However, more research is needed to fully understand the optimal timing and dosing of these combination therapies and to identify biomarkers that can predict which patients are most likely to benefit from this approach [11].

In this work, we focus on VEGFR-2 and PD-L1, which are two promising biological targets for the development of new anticancer drugs due to the fact that they are both key factors in angiogenesis and immune evasion.

VEGFR-2 (vascular endothelial growth factor receptor 2) is a surface protein with an essential function in angiogenesis or neovascularization. Through the inhibition of VEGFR-2, drugs can block angiogenesis and starve tumors of the blood supply they need to grow and metastasize. VEGFR-2 inhibitors, such as bevacizumab and ramucirumab, have shown efficacy in treating multiple cancer types, including colorectal cancer, lung cancer and ovarian cancer. PD-L1 (programmed death-ligand 1) is a surface protein overproduced in some cancer cells, and when it binds to PD-1 (programmed cell death protein 1) on T-cells, these become unable to destroy tumor cells. By blocking the interaction between PD-L1 and PD-1, drugs enhance the destructive action of the immune system against cancer cells. PD-L1 inhibitors, such as pembrolizumab and nivolumab, have shown efficacy in treating multiple cancer types, including lung cancer, melanoma and bladder cancer [2].

Our research group has been working on the discovery of small molecules that are able to simultaneously block some anticancer targets, such as VEGFR-2 and PD-L1 [12], and to study their effect on the TME [13]. For the design of the structures, we considered the results obtained in our previous studies that describe the action of several sets of aryl urea derivatives, U.I and U.II, bearing a styryl moiety (see Figure 1). We found that the halophenyl urea unit is one of these scaffolds that lead to promising small molecule immunomodulator agents due to their multitarget action [14,15,16]. Some small molecules bearing a urea unit have already been described as PD-L1 inhibitors, such as urea CA-170, which was developed by Aurigene, and antiangiogenic compounds, such as sorafenib [17]. In addition, Bristol–Meyers–Squibb developed PD-L1 inhibitors bearing a biphenyl unit linked to a further aromatic ring through a benzyl ether bond (for example, BMS-8, as structure in Figure 1) [17]. With all of this information to hand, we decided to develop some new derivatives, generically labelled as U.III in Figure 1, bearing an aryl urea moiety connected to another aromatic group by a more extensive and flexible chain through the intermediacy of a propenyl functionality. Here, we present the synthesis and the biological study of these new U.III derivatives (see Figure 1 for their specific structures), including their effect on immune cells.

## 2. Results and Discussion

### 2.1. Synthetic Strategy for the Obtention of Urea-Bearing Compounds

The synthesis of the 1,3-diphenylpropenyl aryl ureas began with the preparation of 3-(3-(4-methoxyphenyl)propyl)aniline **C.20** (see Figure 1). Thus, (4-methoxyphenyl)ethan-1-ol **C.15** was converted into 1-(2-bromoethyl)-4-methoxybenzene **C.16** upon reaction with PBr_3_. The **C.16** treatment with triphenylphosphine afforded the phosphonium salt **C.17**, which upon Wittig reaction with 3-nitrobenzaldehyde **C.18** afforded 1-(3-(4-methoxyphenyl)prop-1-en-1-il)-3-nitrobenzene **C.19** as an *E/Z* mixture, which in turn was converted into aniline **C.20** through hydrogenation.

Aniline **C.20** was used to obtain the desired ureas **C.2**–**C.14** and carbamate **C.1**. As far as the latter is concerned, it was obtained upon the reaction of aniline **C.20** with phenyl chloroformate **C.21**. Finally, the desired ureas were synthesized through the reaction of the corresponding anilines **C.22**–**C.34** with triphosgene followed by the addition of compound **C.20** to the reaction mixture.

### 2.2. Biological Evaluation

#### 2.2.1. Cell Proliferation Inhibition

The action on cell proliferation caused by our developed compounds was studied using an MTT assay using the human tumor cell lines of HT-29 (colon adenocarcinoma) and A-549 (pulmonary adenocarcinoma), as well as toward the non-tumor cell line HEK-293 (human embryonic kidney cells), Jurkat T-cells and human microvessel endothelial cells (HMEC-1). This assay allowed us to establish the corresponding IC_50_ values (expressed as the concentration in μM, at which 50% of cell viability is achieved) after 48 h of treatment, which are shown in Table 1, in which the IC_50_ values for the reference compounds sorafenib and BMS-8 are also included.

In general, the tested compounds were more active against HT-29 than A-549 (see Table 1). IC_50_ values were in the micromolar range except for compounds **C.1** (carbamate), **C.3** (*p*-fluorophenyl urea), **C.4** (*m*-fluorophenyl urea), **C.12** (*p*-methoxyphenyl urea) and **C.14** (*o*-methoxyphenyl urea) that were above 100 micromolar on either HT-29, A-549 or HEK-293 while **C.2** (phenyl urea) and **C.8** (*o*-chlorophenyl urea) were also no effective inhibiting A-549 cell proliferation. These results are similar to the ones obtained for the reference compounds.

On the other hand, there was no inhibition of the proliferation of Jurkat T cells or HMEC-1 except for **C.7** (*m*-fluorophenyl urea), **C.10** (*m*-bromophenyl urea), **C.11** (*o*-bromophenyl urea) and **C.13** (*m*-methoxyphenyl urea) with IC_50_ values of around 20 μM.

Finally, we established the tumor-selectivity indexes (SI, see Table 2) for the compounds and these were calculated by dividing the IC_50_ mean against normal cells by the IC_50_ mean against tumor cells. Selectivity indexes below 1 mean poor selectivity toward cancer cells in their inhibitory or cytotoxic effect. According to these data, compounds **C.1** (carbamate), **C.3** (*p*-fluorophenyl urea), **C.12** (*p*-methoxyphenyl urea) and **C.14** (*o*-methoxyphenyl urea), with no inhibitory effect toward any cell line were selected for further biological studies.

#### 2.2.2. Effect on Cellular PD-L1 and VEGFR-2 in Cancer Cell Lines

In our previous studies, we evaluated the action of some ureas on PD-L1 and VEGFR-2 proteins, and we found that these ureas were more active on colorectal cancer cell line HT-29 [14,15]. Thus, we decided to assess the effect of our new ureas on these two proteins only on the HT-29 cancer cell line through the use of flow cytometry.

We studied surface and total PD-L1, and the surface and total VEGFR-2 were relatively determined using DMSO-treated cells as a control and sorafenib and BMS-8 as a positive one. The cells were incubated for a period of time of one day with doses of each of the selected compounds at two different doses: 20 and 100 µM concentrations.

As the tested compounds did not show a significant effect on membrane PD-L1 or VEGFR-2, Table 3 only shows the effect on total PD-L1 and VEGFR-2. No significant effect was achieved for sorafenib at either of the tested doses, while BMS-8 inhibited PD-L1 expression in a dose-dependent manner; that is, it was 40% at 100 μM and around 15% at 20 μM.

Due to the good results obtained for inhibition on both targets, we used the same selected compounds for further studies to determine their potential action as antiangiogenic and immunomodulatory agents.

#### 2.2.3. Study of the Action on Microvessel Formation on Matrigel

The antiangiogenic action of the selected compounds, as well as the reference compounds, was evaluated by determining the effect on the formation of new microvessels formed by HMEC-1 endothelial cells on a Matrigel matrix. Table 4 shows the lower dose at which these compounds are able to inhibit microtube formation.

Figure 2 displays the pictures of the inhibition of neovascularization achieved by compounds **C.1** and **C.12** at different concentrations.

#### 2.2.4. Effect on Cancer Cell Proliferation in Co-Cultures with Monocytes THP-1

To establish the potential of the selected compounds as immunomodulator agents, we studied the effect of the selected compounds on HT-29 cell proliferation when co-cultured with human monocytic leukemia cell line THP-1. For the study, we used different proportions of cancer and immune cells. Standard protocol proportions were 1:5 cancer/immune cells. We include a test using a 2:1 proportion of HT-29 cells to THP-1 cells. The tests were performed for two periods of time: 24 and 48 h, using 100 μM doses of the selected compounds and BMS-8 as the reference compound.

The results shown in Table 5 show that the effect on HT-29 cell proliferation was higher after 48 h than after 24 h and did not depend on the proportion of cancer and immune cells.

Figure 3 shows the morphological changes suffered by HT-29 cells after 48 h of co-culture with THP-1 at a (1:5) proportion. We observed that the control cells preserved a morphology related to the epithelial nature of HT-29 cells, while when they were treated with BMS-8 and the selected derivatives, the cells retained this epithelial nature though increased and brighter cytosolic granulation appeared.

#### 2.2.5. Effect on Immune Cell Proliferation in Co-Cultures of HT-29/THP-1

We also determined the effect of the selected compounds on the human monocyte cells THP-1 in the described co-cultures (see Table 6). In general, none of the compounds had a significant effect on immune cell viability.

It has been demonstrated that cancer, and all the therapies associated with this illness, promote functional alterations in monocytes, such as the acquisition of immunosuppressive activity in TME, which is related to the expression of CD11b, an integrin, which, when binding to CD18, promotes increased invasiveness and metastasis of tumor cells [18]. Reducing the expression of CD11b has become a promising target for immune modulation in anticancer therapies. For that reason, we decided to study the effect of our compounds on CD11b in THP-1 cells co-cultured with HT-29. We also determined the relative amount of CD80, a common surface marker for monocytes.

Table 7 shows the most significant results obtained for the expression of CD80 and CD11b proteins on the THP-1 membrane. The results are related to non-treated THP-1 cells when they were co-cultured with HT-29.

## 3. Discussion

We have synthesized thirteen propenylureas and one carbamate to determine their capability as potential multitarget inhibitors of VEGFR-2 and PD-L1 proteins related to the immunosuppressant activity of TME.

In terms of the effect on cancer cell lines, in general, the synthetic compounds are more active against HT-29 than A-549 (see Table 1). We found IC_50_ values in the range of micromolar comparable to that shown by reference compounds sorafenib and BMS-8, except for compounds **C.1** (carbamate), **C.3** (*p*-fluorophenyl urea), **C.4** (*m*-fluorophenyl urea), **C.12** (*p*-methoxyphenyl urea) and **C.14** (*o*-methoxyphenyl urea) that have no effect on either HT-29 or A-549, while **C.2** (phenyl urea) and **C.8** (*o*-chlorophenyl urea) did not effectively inhibit A-549 cell proliferation.

Regarding the inhibitory effect on the non-cancer cell line HEK-293, the tested compounds behave in a similar way to HT-29, showing moderate IC_50_ values except for **C.1** (carbamate), **C.3** (*p*-fluorophenyl urea), **C.12** (*p*-methoxyphenyl urea) and **C.14** (*o*-methoxyphenyl urea), which exhibited IC_50_ values above 100 μM.

In addition, we observed that all of the compounds exhibited similar effects on Jurkat T cells as HMEC-1; that is, they did not effectively inhibit the proliferation of these cells, except for **C.7** (*m*-fluorophenyl urea), **C.10** (*m*-bromophenyl urea), **C.11** (*o*-bromophenyl urea) and **C.13** (*m*-methoxyphenyl urea), with IC_50_ values of around 20 μM.

We also observed that most of the compounds were found to be selective towards cancer cells as their IC_50_ values and, what is the same, their inhibitory effect on cell proliferation was significantly higher on the tested cancer cell lines of HT-29 and A-549 than on the immune and endothelial cells and Jurkat T and HMEC-1 cells, respectively. Compared to the effect on non-cancer cells, HEK-293, some compounds exhibited selective inhibitory action against cancer cell proliferation at lower doses than the ones against HEK-293, yielding selective indexes above 1 (see Table 2) [19]. In addition, compounds **C.1**, **C.3**, **C.12** and **C.14** exhibited no antiproliferative action in either of the tested cell lines.

From the observations provided, it can be concluded that there is a relationship between the structure of the synthetic compounds and their antiproliferative activity. Thus, carbamate and fluoro and methoxy phenyl ureas were the less toxic compounds, while for the rest of the ureas, *p*-aryl-substituted ones were more active in the inhibition of cancer cell proliferation than the *m*-substituted ones, and these proved to be more active than the *o*-substituted ureas.

The compounds with no inhibitory effect on any cell line (**C.1**, **C.3**, **C.12** and **C.14**) were chosen for further biological studies in order to determine their potential as onco-immunomodulatory agents.

From these studies, we found that some of the selected compounds exhibited significant inhibitory effects on both total PD-L1 and VEGFR-2 on the HT-29 cell line.

In general, the effect of the selected compounds was not dose-dependent. In fact, at 20 μM, the compounds were more active than at 100 μM. The most active derivative as a dual inhibitor was **C.12** (*p*-methoxyphenyl urea), showing inhibition rates of around 75% for both PD-L1 and VEGFR-2. **C.14** (*o*-methoxyphenyl urea) yielded a 50% inhibition of PD-L1, while **C.3** (*p*-fluorophenyl urea) and **C.1** (carbamate) inhibited 30% of PD-L1 and about 45 % of VEGFR-2. With these observations, we could conclude that *p*-substituted ureas, such as **C.3** and **C.12**, were the most active ones in inhibiting the studied targets, yielding more than 50% inhibition rates. Moreover, the same relationship was found between structure and antiangiogenic properties as, again *p*-sustituted ureas, as **C.3** and **C.12**, were the most active in preventing the formation of new microvessels on matrigel HMEC-1 cell cultures. Moreover, if we compare the values of minimum active concentration with IC_50_ values for the HMEC-1 cell line, we find that the selected compounds exhibited antiangiogenic action while having no effect on endothelial cell proliferation.

The selected compounds were also tested for their effect on cancer cell proliferation and immune cell viability in co-culture experiments using HT-29 and THP-1 cells. From this study, we established that the effect of the tested compounds on cancer cell viability in the presence of THP-1 was more prominent after 48 h of treatment, though at 24 h, important morphological changes had already been produced, which were being affected by the proportion of cancer and immune cells. In addition, the effect of the compounds was no-dose dependent. **C.1** (carbamate), and again *p*-substituted ureas were the most active ones, as **C.3** (*p*-fluorophenyl urea) and **C.12** (*p*-methoxyphenyl urea), showing 50–45% inhibition of cancer cell viability at both (1:5) and (2:1) proportions after 48 h.

On the other hand, we also observed that the compounds caused morphological changes in HT-29 cells after 48 h of co-culture with THP-1 at a (1:5) proportion. While the control cells preserved a morphology related to the epithelial nature of HT-29 cells [20], we found that, when the co-cultures were treated with BMS-8 and the selected derivatives, the cells retained this epithelial nature though increased and brighter cytosolic granulation appeared. In addition, these treatments led also to a slight loss of cell-to-cell contact and the appearance of clustered cells with round-shaped cells and irregular surfaces [21], together with a less adhesive rounded morphology that resulted in cell scattering with apoptotic features [22]. In addition, derivatives **C.1** and **C.3** led to less confluent cultures. All of these are morphological changes related to the loss of their epithelial appearance by the synergic action of the compounds and immune cells, while none of the tested compounds had any effect on the immune cell viability.

Finally, the compounds were tested for their effect on CD11b and CD80 expression in THP-1 cells co-cultured with HT-29. While the effect on CD80 expression was very mild (less than 10% of inhibition rates), all of the compounds tested were found to significantly reduce CD11b expression, which is a promising target for immunomodulation in anticancer therapies [23,24]. In most of the cases, this effect was even higher than the one observed for the reference compound BMS-8. In this sense, **C.3** (*p*-fluorophenyl urea) and **C.14** (*o*-methoxyphenyl urea) reduced about 35% of CD11b, while **C.12** (*p*-methoxyphenyl urea) and carbamate **C.1** inhibited around 30% the expression of CD11b.

## 4. Materials and Methods

### 4.1. Symthetic Protocols

#### 4.1.1. General Techniques

High-resolution mass spectra were taken with an electrospray ionization–mass spectrometer, ESI–MS. Nuclear Magnetic Resonance spectra were analyzed at room temperature (25 °C), and the references taken for the peaks were the ones that came from the solvents. The IR spectra were analyzed using NaCl pills. We only describe the peaks for the most important functional groups. Commercially available reagents were used as received. We used N2 for the inert conditions when they were needed.

#### 4.1.2. Experimental Procedure for the Synthesis of Ureas **C.2**–**C.14**

A solution of the corresponding aniline (1.0 mmol) dissolved in dry THF (5.0 mL) was slowly dripped into a stirred solution of triphosgene (303 mg, 1.0 mmol) in dry THF (5.0 mL). Then, Et_3_N (279 µL, 2.0 mmol) was slowly added to the reaction mixture and the resulting mixture was stirred at room temperature for 1 h and then refluxed for 5 h under a nitrogen atmosphere. After this time, the reaction mixture was cooled to room temperature and the solid was filtered off. After the evaporation of the solvent in vacuo, the residue was taken up in dry THF (5.0 mL), and a THF solution (5.0 mL) of 3-(3-(4-methoxyphenyl)propyl)aniline (**C.20**) (241 mg, 1.0 mmol) was directly added, followed by the addition of Et_3_N (139 µL, 1.0 mmol). The resulting mixture was refluxed under a nitrogen atmosphere overnight. Then, the solvent was removed in vacuo, and AcOEt (20 mL) was added. The organic phase was washed with aqueous HCl 10% (2 × 20 mL) and brine and dried over Na_2_SO_4_. Then, the solvent was removed in vacuo, and the residue was recrystallized from acetonitrile and dried under vacuum to yield ureas **C.2**–**C.14** as white solids (34–92%) (Appendix A).

### 4.2. Biological Studies

#### 4.2.1. Cell Culture

The cell media were obtained from Gibco (Grand Island, NY, USA). Fetal bovine serum (FBS) came from Harlan-Seralab (Belton, UK). The rest of the supplements or chemicals not listed in this section were obtained from Sigma Chemical Co. (St. Louis, MO, USA). The plastics used for cell culture were supplied by Thermo Scientific BioLite (Waltham, MA, USA). We used an IBIDI μ-slide angiogenesis (IBIDI, Martinsried, Germany) for the antiangiogenic test. Stock solutions of the compounds were in DMSO (20 mM) and preserved at −20 °C.

HT-29, A549, HEK-293 and Jurkat cell lines were cultured in Dulbecco’s modified Eagle medium (DMEM) containing glucose (1 g/L), glutamine (2 mM), penicillin (50 μg/mL), streptomycin (50 μg/mL) and amphotericin B (1.25 μg/mL) supplemented with 10% FBS. For the HMEC-1 cell line, we used Dulbecco’s modified Eagle medium (DMEM)/Low glucose containing glutamine (2 mM), penicillin (50 μg/mL), streptomycin (50 μg/mL, and amphotericin B (1.25 μg/mL) supplemented with 10% FBS. For the development of the antiangiogenesis test, the HMEC-1 cells were seeded on matrigel in EGM-2MV Medium supplemented with EGM-2MV SingleQuots (Lonza, CA, USA).

#### 4.2.2. Cell Proliferation Assay

In general, 5 × 10^3^ (HT-29, A549, HMEC-1, Jurkat and HEK-293) cells per well were seeded in 96-well plates with 1:1 dilutions of the compounds in a volume of 100 μL of the cell media. The 3-(4,5-dimethylthiazol-2-yl)-2,5-diphenyltetrazolium bromide (MTT; Sigma Chemical Co.) dye exclusion assay was performed, and after 48 h (37 °C, 5% CO_2_ in a humid atmosphere), 10 μL of MTT (5 mg/mL in phosphate-buffered saline, PBS) was added to each well and incubated for 3 h at 37 °C. After the supernatant was discarded, we solved the formazan crystal in 100 µL of DMSO. The absorbance was then registered at 550 nm using a plate lector. For all concentrations of any compound, cell viability was expressed as the percentage of the ratio between the mean absorbance of the treated cells and the mean absorbance of the untreated cells. Three to five experiments were performed to establish the IC_50_ values (i.e., concentration half inhibiting cell proliferation, we used GraphPad Prism 7 software).

#### 4.2.3. PD-L1 and VEGFR-2 Relative Quantification by Flow Cytometry

The effects of the compounds on the target in the cancer cell lines compounds were tested using 20 and 100 µM doses. For that, 10^5^ cells per well were seeded on a 12-well plate for 24 h with the corresponding dose of the tested compound in a total volume of 500 μL of cell media. To determine the total PD-L1 and VEGFR-2, the cells were collected and fixed with 4% in a PBS of formaldehyde. After this, a treatment with 0.5% in PBS Triton^TM^ X-100 permeabilized the membranes. Finally, the cells were treated with FITC Mouse monoclonal Anti-Human VEGFR-2 (ab184903) and Alexa Fluor^®^ 647 Rabbit monoclonal Anti-PD-L1 (ab215251).

To determine surface PD-L1 and VEGFR-2, the assay used was similar to the above but avoided the permeabilization step (Triton^TM^ X-100).

#### 4.2.4. Microvessel Formation Inhibition Assay

We used IBIDI μ-slide angiogenesis plates (IBIDI, Martins ried, Germany). The wells were coated with 15 μL of Matrigel^®^ (10 mg/mL, BD Biosciences, San Jose, CA, USA) at 4 °C and incubated at 37 °C for at least 30 min. Afterward, the HMEC-1 cells were seeded at 2 × 10^4^ cells/well in 25 μL of medium and were left for 30 min at 37 °C while they attached. Then, the compounds were added in serial doses in 25 μL of the medium, and after several hours (maximum 24 h) of incubation at 37 °C, pictures for the wells were taken to evaluate the formation of microvessels.

#### 4.2.5. Cancer and Immune Cell Proliferation Test in Co-Cultures

To determine the action of the compounds on cancer cell proliferation in co-culture with immune cells, 10^5^ or 2 × 10^5^ of the HT-29 cells line were seeded in each well of 12-well plates and left at 37 °C for 24 h. Then, the cancer cells were treated with IFN-γ (10 ng/mL; human, Invitrogen^®^, Waltham, MA, USA) containing 5 × 10^5^ or 10^5^, respectively, of THP-1 cells per well and the corresponding compound at 100 μM or DMSO for the positive control. After 24 h/48 h of incubation, the supernatants were collected to establish the alive THP-1 cells using FSC and SCC flow cytometry. In addition, cancer cells were collected with trypsin, fixed with 4% in PBS paraformaldehyde and counted using flow cytometry.

#### 4.2.6. CD11b and CD80 Detection THP-1

The action of the compounds on CDs in co-cultured THP-1 immune cells with cancer cell line HT-29 was tested by treating them with the compounds for 24 h/48 h (see Section 4.2.5). Surface CD11b and CD80 was determined by collecting THP-1 with trypsin, fixing with formaldehyde and staining with FITC Mouse monoclonal Anti-Human CD80 (Sigma-Aldrich SAB4700142) and Alexa Fluor^®^ 647 Rabbit monoclonal Anti-CD11b (Merck #MABF366, Rahway, NJ, USA).

## 5. Conclusions

The study described in this paper provides valuable insights into the potential of small molecules bearing benzylethylenearyl–urea moieties. 

We have established that depending on the aryl substituent and relative position in the ring, we can modulate the inhibitory effect on both total PD-L1 and VEGFR-2 on the HT-29 cell line in no dose-dependent manner and at non-cytotoxic concentrations of compound. Compounds **C.3** (*p*-fluorophenyl urea) and **C.12** (*p*-methoxyphenyl urea) were the most promising agents for several reasons: they inhibited 50% of both biological targets; they were the most effective as antiangiogenic agents: they reduced to the half the cancer cell viability in the presence of monocytes and reduced CD11b expression on monocytes without causing any damage on them. All these biological actions of these compounds suggest that they are promising agents for immunomodulation in anticancer therapies and they launch benzylethylenearyl–urea as good scaffolds for the design of new anticancer drugs. Further studies are needed to determine their efficacy and safety in preclinical settings.

## Data Availability

Data is contained in within the article or Appendix A.

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
