# Peer review of "Unveiling the Potential of BenzylethyleneAryl–Urea Scaffolds for the Design of New Onco Immunomodulating Agents"

_pharmaceuticals, 2023, doi:10.3390/ph16060808_

Round 1
Reviewer 1 Report
The paper by Gil-Edo and coauthors describes the development of a series urea-based compounds as onco-immunomodulating agents. The the data are in general sound, so these compounds could be useful as lead compounds for further development.
My big concern is about the data in the Supporting Information:
1) Dose-response curves of the antiproliferative effects determined by MTT assay are completely missing;
2) 1H and 13C NMR have to be changed in 1H and 13C NMR
3) Integration of signals in the proton NMR are required, otherwise is difficult to check the spectrum
4) IR spectrum has extremely low quality, pickpicking is missing
5) HPLC chromatogram at least of the selected compounds is necessary for proving the purity of the molecules
Author Response
Please find answers in the attached file

Reviewer 2 Report

Minor English language correction required
Author Response

(The authors gave the same response as above.)

Round 2
Reviewer 1 Report
Thank you for the answers and corrections